# Performance of amplicon and capture based next-generation sequencing approaches for the epidemiological surveillance of Omicron SARS-CoV-2 and other variants of concern

Carlos Daviña-Núñez[1,2], Sonia Pérez[1,3]*, Jorge Julio Cabrera-Alvargonzález[1,3], Anniris Rincón-Quintero[1,3], Ana Treinta-Álvarez[3], Montse Godoy-Diz[3], Silvia Suárez-Luque[4], Benito Regueiro-García[1]

1 Microbiology and Infectology Research Group, Galicia Sur Health Research Institute (IIS Galicia Sur), Vigo, Spain, 2 Universidade de Vigo, Vigo, Spain, 3 Microbiology Department, Complexo Hospitalario Universitario de Vigo (CHUVI), SERGAS, Vigo, Spain, 4 Dirección Xeral de Saúde Pública, Xunta de Galicia, Consellería de Sanidade, Santiago de Compostela, A Coruña, Spain

* sonia.maria.perez.castro@sergas.es

**Data Availability Statement:** The genetic sequences analysed in the publication are

## Abstract

To control the SARS-CoV-2 pandemic, healthcare systems have focused on ramping up their capacity for epidemiological surveillance through viral whole genome sequencing. In this paper, we tested the performance of two protocols of SARS-CoV-2 nucleic acid enrichment, an amplicon enrichment using different versions of the ARTIC primer panel and a hybrid-capture method using KAPA RNA Hypercap. We focused on the challenge of the Omicron variant sequencing, the advantages of automated library preparation and the influence of the bioinformatic analysis in the final consensus sequence. All 94 samples were sequenced using Illumina iSeq 100 and analysed with two bioinformatic pipelines: a custom-made pipeline and an Illumina-owned pipeline. We were unsuccessful in sequencing six samples using the capture enrichment due to low reads. On the other hand, amplicon dropout and mispriming caused the loss of mutation *G21987A* and the erroneous addition of mutation *T15521A* respectively using amplicon enrichment. Overall, we found high sequence agreement regardless of method of enrichment, bioinformatic pipeline or the use of automation for library preparation in eight different SARS-CoV-2 variants. Automation and the use of a simple app for bioinformatic analysis can simplify the genotyping process, making it available for more diagnostic facilities and increasing global vigilance.

## Introduction

SARS-CoV-2, a novel betacoronavirus, was first identified in Wuhan, China, in December 2019. The virus was associated with an increase of cases of a novel pneumonia later defined as coronavirus disease 2019 (COVID-19). Detection and characterization of cases have been amongst the governmental efforts around the world to control the spread of the SARS-CoV-2

published in the database epiCoV from GISAID. All GISAID IDs for each sequence can be found in the Supporting Information files.

**Funding:** This work has been funded by: the European Centre for disease Prevention and Control under the GA ECDC/HERA/2021/024 ECD.12241, the Aid for the consolidation and structuring of competitive research units and other promotion actions of the Galician Innovation Agency, code IN607B-2022/19, and the Consellería de Sanidade, Galicia, Spain The funders had no role in study design, data collection and analysis, decision to publish, or preparation of the manuscript.

**Competing interests:** The authors have declared that no competing interests exist.

pandemic. In order to do so, healthcare systems have focused on ramping up their capacity for diagnosis throughout RT-PCR testing as well as for epidemiological surveillance through viral whole genome sequencing (WGS). For example, SARS-CoV-2 sequences uploaded to the GISAID database went from 313k in 2020 to 6.36M in 2021 and 7.75M in 2022 [1]. Across this time, several variants of concern (VOC) and variants of interest have become predominant throughout the world given their ability to spread faster or to avoid the immune system [2]. WGS data provided relevant information for SARS-CoV-2 circulating clusters, vaccine development or even insights on the intermediate zoonotic hosts for SARS-CoV-2 [3].

High-throughput next-generation sequencing (NGS) allows for massive parallel sequencing of DNA fragments. Viral WGS from clinical samples through NGS usually requires a step of viral nucleic acid enrichment in order to increase the sequencing yield. While unbiased metagenomic NGS without enrichment is possible as a mechanism for viral sequencing, it requires a very high number of reads per sample to obtain sufficient viral reads. It is a suboptimal approach, especially considering the NGS reagents cost [4]. A previous study found metagenomic approaches to map below 6% of the total reads to SARS-CoV-2, with a majority of reads being host DNA [5].The most common methods of viral enrichment are the amplicon-based and the capture-based approaches [6, 7]. In a nutshell, the amplicon-based methods rely on PCR to amplify viral genomic material using specific primers to cover the target region, while the capture-based methods use specific oligos that hybridise to the target regions, followed by a purification of the oligo-bound target DNA.

Despite the potential benefits of viral sequencing, there are challenges to the NGS implementation in diagnostic facilities. Firstly, viral enrichment and sequencing-ready library synthesis require a high degree of expertise. Secondly, although sequencing has become more affordable with the new NGS technologies, the overall cost is high, and the spending must be justified in order to implement automated high-throughput sequencing and departments that regularly track viral variants. Obtaining all the necessary laboratory equipment can be challenging as well, as there is a need for space and resources for flow chambers, freezers, sequencers, etc. Finally, bioinformatic analysis is another barrier to overcome, as it requires a skilled responsible or an user-friendly pipeline, which is not always available.

A way to reduce complexity and chances of contamination is the automation of the library preparation steps, especially when working with a high volume of samples. Commercially available pipetting platforms can integrate both enrichment and library preparation in the same workflow, reducing hands-on time [8, 9].

In summary, simple, cost-effective, high-throughput protocols of viral enrichment and library preparation, together with user-friendly online bioinformatic analysis tools, would make sequencing of SARS-CoV-2 more accessible to sequencing facilities, even in locations with more moderate resources.

In this paper, we tested the performance of two protocols of viral nucleic acid enrichment available for SARS-CoV-2. We selected the Illumina iSeq 100 platform, the smallest and most affordable Illumina sequencers, because of its ability to yield the fastest results. We also focused on the challenge of the Omicron variant for sequencing, the advantages of automated library preparation and the influence of the bioinformatic analysis in the final sequence generation.

## Materials and methods

### Sample selection

Nasopharyngeal swab samples from patients were selected from RT-PCR confirmed SARS-CoV-2 cases in the area of Vigo, a city in Northwest Spain. Swabs were transported in Vircell transport medium (Vircell, Granada, Spain) and frozen until viral RNA extraction. Sample

cycle threshold (CT) ranged from 8 to 26. A first cohort was composed of fifty-four samples with different SARS-CoV-2 variants collected from March to June, 2021. A second cohort was added with forty Omicron samples collected from May to July, 2022. No positive or negative controls were added to the NGS runs.

## Nucleic acid extraction

In the first cohort, for the amplicon-based enrichment (ABE) approach, MagNAPure 24 Total NA isolation kit (Roche Diagnostics, Mannheim, Germany) was used for RNA extraction. For the KAPA capture-based enrichment (CBE) approach, RNA extraction was performed from the same nasopharyngeal samples using the QIAGEN QiaAmp DNA Mini kit in a QiaCube extractor (Qiagen, Hilden, Germany).

In the second cohort, RNA was extracted using QIASymphony DSP Virus Pathogen Midi kit (Qiagen) according to manufacturer's instructions.

## Sequencing approaches

**Amplicon-based enrichment—manual library preparation.** For the first cohort, the reverse transcription was performed with random hexamers (Invitrogen, California, USA) and SuperScript™ IV kit (SSIV) (Invitrogen, California, USA). The amplification was performed with the ARTIC v3 primer panel (Integrated DNA Technologies, California, USA), a set of 98 primer pairs divided into two pools, enough to cover the whole genome (S1 Table) and the Q5 TaqPolymerase kit (New England Biolabs, Massachusetts, USA), as previously described [10]. The detailed manual RT-PCR protocol is found in S1 File. Enriched samples were then normalised and libraries were prepared using the Illumina DNA prep kit (Illumina Inc., California, USA) according to the manufacturer's instructions. Clean-up of libraries was performed using Ampure XP beads (Beckman Coulter, California, USA) in a 1.8:1 beads-to-sample ratio.

**Amplicon-based enrichment—automatic library preparation.** For the second cohort, samples were enriched using the ARTIC v4.1 primer panel, an updated version for optimal amplification of the Omicron variant (S1 Table) [11]. Retrotranscription, enrichment and library preparation were performed using the Illumina CovidSeq test (Illumina, Protocol in Illumina Document #1000000126053 v04). All steps of the Illumina CovidSeq protocol were performed according to the manufacturer's instructions by the HAMILTON Microlab STAR pipetting platform (Hamilton Iberia, Barcelona, Spain).

**Capture-based enrichment.** Capture-based libraries were prepared following the KAPA RNA HyperCap workflow with specific enrichment probes for SARS-CoV-2 (Roche Diagnostics, Mannheim, Germany). Each individual library was created using 10ul of extracted RNA input and following the protocol established by the kit's manufacturer. RNA was fragmented at 96° for 6 minutes. Eighteen PCR cycles were used to enrich each library prior to capture. Libraries were quantified and then multiplexed in sets of 6 libraries. For a total of 1500 ngs of DNA per capture, 250 ngs per library were pooled. Captured pools were amplified using 17 PCR cycles and then quantified for sequencing.

**Genomic library analysis.** Genomic libraries were sequenced using an Illumina iSeq 100 with 2x151 paired-end cycles. A total of 18 or 20 samples per run were sequenced in the first and second cohort, respectively. Sample pools were diluted to 75 pM and added into an iSeq cartridge v2 with Illumina PhiX at 5% concentration.

All libraries enriched with an ABE approach were quantified using Qubit dsDNA HS Assay Kit (Thermo Fisher Scientific, Massachusetts, USA). The libraries obtained from the CBE approach were quantified using the qPCR KAPA library quantification kit (Roche Diagnostics,

Cape Town, South Africa). Sample size of all libraries was checked using an Agilent 2100 Bioanalyzer (Agilent technologies, California, USA) prior to sequencing.

**Bioinformatic analysis.** The quality of the fastq files was checked with FastQC 0.11.9 (Andrews 2010) and QualiMap 2.2.1 [12]. The reads were aligned to the reference NCBI code NC_045512.2 from Wuhan with BWA-mem2 [13] w. We used iVar 1.3 [14] to trim primer sequences and the reads based on a quality threshold (Default: 20) and to remove reads less than 32 bp ong. We used SAMtools v1.10 *coverage* (using htslib 1.10.2) to calculate the genome coverage [15].

To build a consensus sequence for each sample, we merged the reads with SAMtools *mpileup* and used iVar 1.3 [14] consensus with a minimum quality score threshold to count base of 20, a minimum read depth of 10 to call consensus and a minimum VAF threshold of 0.01. We assigned the consensus sequences to a SARS-CoV-2 clade with Nextclade [16] and to a SARS-CoV-2 PANGO lineage [17] with Pangolin [18].

We considered the ECDC recommendations to establish a quality threshold (QC) for each SARS CoV-2 consensus sequence: Minimum read depth of 10 over at least 95% of the genome [19]. As an additional quality threshold, samples with a median base depth below 50 were discarded due to low sequencing quality and were discarded from further analysis.

The Illumina-owned DRAGEN™ Covid Lineage 3.5.6, using 10X as coverage threshold for base-calling, was also used. DRAGEN™ (Dynamic Read Analysis forGENomics) is a Bio-IT Platform in BaseSpace™ Sequence Hub. SARS-CoV-2 variant calling was performed using Nextclade [16], based on the consensus sequences generated.

All statistical data analysis was done using R (version 4.1.1, https://cran.r-project.org/). Shapiro-Wilk normality test was performed to check for normality. Wilcoxon-sign rank sum test and Fisher's F-test were used. A p-value below 0.05 was considered significant.

Multiple sequence alignment was performed using Multiple Alignment using Fast Fourier Transform (MAFFT) [20] and the aligned sequences were used to generate Neighbour-Joining phylogenetic trees in MEGA11 [21] with the Maximum Composite Likelihood model. For the consensus sequence comparison, unread bases and terminal bases were excluded from the mismatch count. Data visualisation was performed with the R program ggplot2 [22].

## Results

Samples from nasopharyngeal swabs were tested from two Cohorts, a pre-Omicron cohort (n = 54) and an Omicron cohort (n = 40). All samples were enriched using both methods, an amplicon-based method (ARTIC v3 and Illumina DNA prep for Cohort 1, ARTIC v4.1 and Illumina CovidSeq for cohort 2) and a capture-based method (KAPA RNA Hypercap). Low to mid CT value samples were chosen (8–26). Read depth, genome coverage, allele frequency and consensus sequences were analysed in order to evaluate the yield of both methods. From 94 samples, 6 did not pass QC for sequencing using the capture-based method, obtaining 88 correctly sequenced samples.

### SARS-CoV-2 base coverage

Median base depth over 50 was obtained for 100% (94/94) of the samples with the ABE and 94% (88/94) using the CBE (S2 Table). Among these samples, ABE showed a higher median base depth than CBE (median ± standard deviation (SD): 1444.5 ± 581 *vs.* 776.5 ± 1426; Wilcoxon test, $p = 0.0057$) [IQR: 933–1894 *vs.* 322–1628]. CBE showed a more heterogeneous depth per sample (Fisher's F-test, $p < 0.0001$). This heterogeneity caused CBE to present the samples with the highest (>5000 reads/base) and the lowest values (<50 reads/base). By Cohort, pre-Omicron samples had a higher read depth than Omicron samples only in the ABE

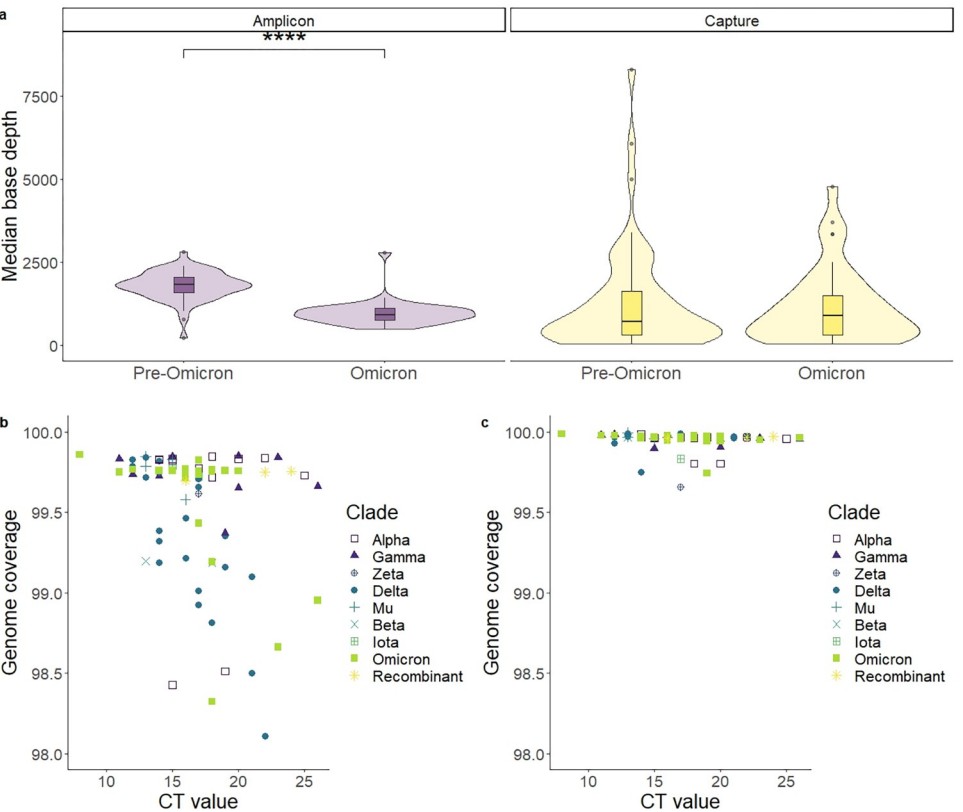

**Fig 1. Overall results of sequencing of SARS-CoV-2 using an amplicon-based method and a capture-based method for enrichment.** a) Violin plot and boxplot of the median read depth divided by cohorts. (For Amplicon enrichment, Wilcoxon test; $p < 0.0001$; For capture enrichment, $p = 0.92$). b,c) Genome coverage, percentage of the genome with a read depth over 10. Each data point corresponds to one sample. Amplicon: ARTIC panel, amplicon-based enrichment. Capture: KAPA RNA HyperCap, capture-based enrichment.

approach (median ± SD: 1841 ± 440 *vs.* 913 ± 378; Wilcoxon test, $p < 0.001$). For the CBE approach, no significant differences were found between both cohorts (median ± SD: 723 ± 1611 *vs.* 909 ± 1141; Wilcoxon test, $p = 0.92$) (Fig 1A).

All samples passing QC had over 98% of genome coverage. The percentage of called bases was higher in the CBE than in the ABE approach (median ± SD: 99.95 ± 0.05 *vs.* 99.56 ± 0.40; Wilcoxon test, $p < 0.0001$) (Fig 1B and 1C).

**Region specific genome coverage.** Non-covered areas by the genome were checked across methods. For the ABE, the number of unread bases was variant-specific with Delta samples having more unread bases than non-Delta samples (mean ± SD: 204 ± 139 *vs.* 107 ± 102, Wilcoxon test, $p = 0.00088$). This correlation was not observed with the CBE, where actually non-Delta samples had less coverage, although with smaller differences (mean ± SD: 12 ± 15 *vs.* 15 ± 16, Wilcoxon test, $p = 0.0026$) (Fig 2A and 2B).

The location of unread bases across the genome was not randomly distributed, but rather concentrated in certain areas of the genome, regardless of the method of enrichment. The ABE showed a maximum of unread bases around base 21850. This peak was Delta-specific (Fig 2C and 2D).

**Variant allele frequency.** We analysed the allele frequency for each SNP detected. In our samples, ABE showed little variance in allele frequency, with most mutations detected with over 90% read agreement (Fig 3A). Using CBE, while most samples showed high allele

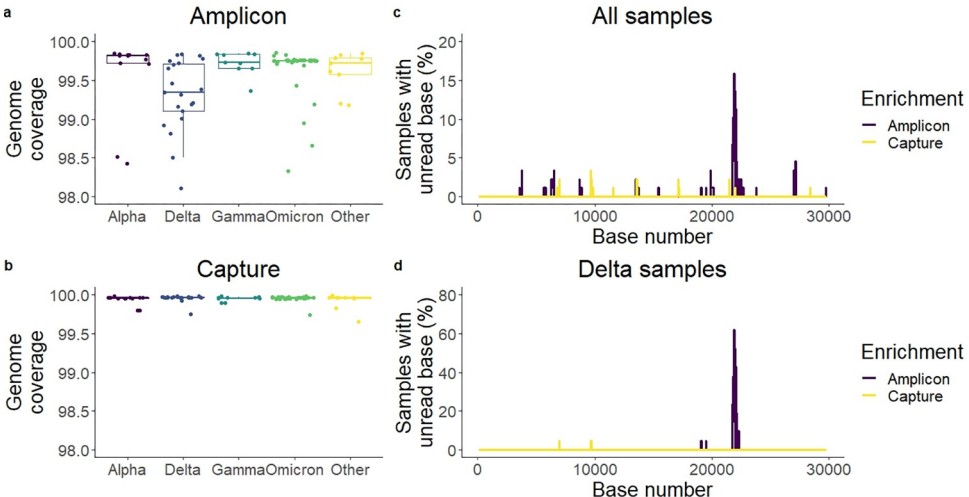

**Fig 2. Evaluation of the effectiveness of two methods of enrichment.** a-b) Genome coverage per variant. Delta samples in the amplicon enrichment showed the lowest genome coverage. c-d) Percentage of samples with less than 10 reads on each base. Areas with dips in coverage were identified, with a notable peak consistent of Delta samples at the beginning of the *spike* gene. Amplicon: ARTIC panel, amplicon-based enrichment. Capture: KAPA RNA HyperCap, capture-based enrichment.

frequency, a subset of samples showed high variability (Fig 3B). Specifically, 6 samples processed with CBE had more than 25 SNPs detected with 20–90% read agreement (low-agreement). For ABE, samples had between 0 and 8 SNPs detected with low agreement (Fig 3C).

**Agreement of consensus sequence across methods.** For all samples but one, the same Pango lineage was determined using both methods. The exception was a sample declared as B.1.1.529 using ABE and BA.2 using CBE (S2 Table). The sample turned out to be a possible

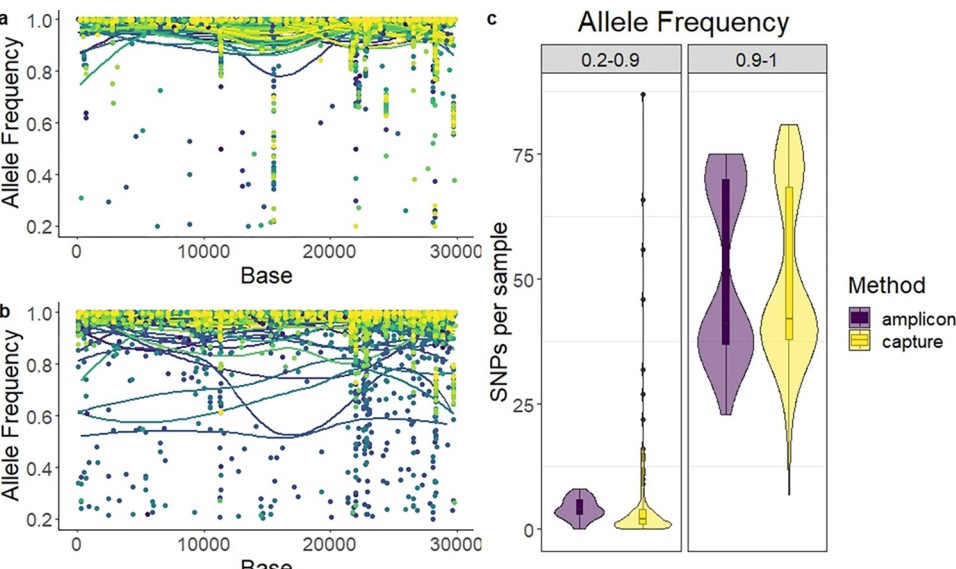

**Fig 3. Allele frequency plot by nucleotide of SARS-CoV-2 genome.** a) ARTIC panel, amplicon-based enrichment. b) KAPA RNA HyperCap, capture-based enrichment. Each point represents an SNP, while each line represents a LOESS local regression for each sample. One colour per sample. c) Violin plot and boxplot of the number of SNPs per sample at low frequency (0.2–0.9) or high frequency (>0.9).

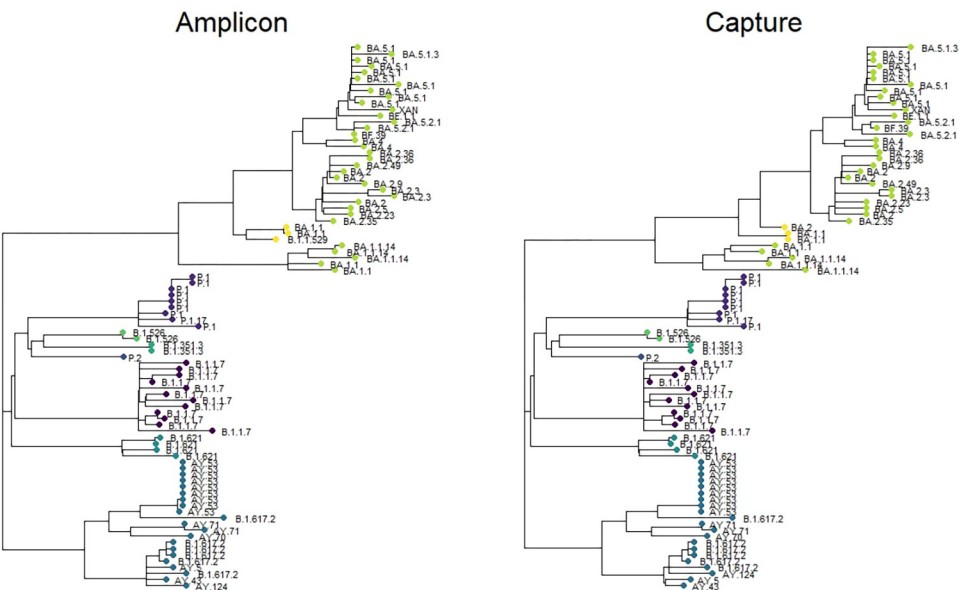

**Fig 4. Phylogenetic tree for all samples analysed in the study.** Samples are coloured by clade, with undesignated recombinant samples shown in yellow. The tip label indicates the Pango lineage. Tree generated by Neighbour-Joining method; Maximum Composite Likelihood. Amplicon: ARTIC panel, amplicon-based enrichment. Capture: KAPA RNA HyperCap, capture-based enrichment.

recombinant between BA.1 and BA.2. Amplicon sequences and capture sequences showed highly similar locations in the phylogenetic tree (Fig 4), although with small branch differences within clusters caused by mismatches detected or by areas left unread due to low coverage.

A total of 84 base changes were observed depending on the method of library enrichment (Fig 5A). From 88 samples, no base mismatch between both methods was observed in 56 samples (64%). A total of 24 (27%), 3 (3%) and 5 (6%) samples showed 1, 2 and more than 2 discrepancies, respectively. The two most common discrepant SNPs were found as errors with the ABE method: The lack of detection of *G21987A* in the Delta samples was the most common mismatch (n = 16). The second most common mismatch was the addition of the mutation *T15521A* in the Omicron samples (n = 12) analysed by the ABE method. Four samples, all analysed by the CBE method, showed a high number of discrepancies (6–24 mismatches), consisting in missing characteristic SNPs.

**Discrepancies associated to variations in filtering parameters and the alignment algorithm.** All consensus sequences were obtained with a customised pipeline, as described in the materials and methods section. A second bioinformatic analysis was performed for all samples using the Illumina® DRAGEN™ COVID lineage app. We compared the consensus sequences generated to evaluate the concordance between a more user-friendly method and our in-house pipeline. In the case of ABE, 17 discrepancies were found, evenly distributed across samples (one per sample) (Fig 5B). The most common of these variations was *T15521A* (n = 5), which had an allele frequency between 0.22 and 0.93 in ABE (Fig 3A). For CBE, 39 mismatches were found concentrated in nine samples, with 79 out of 88 samples having no discrepancies (Fig 5C). These 9 samples showed a low quality of sequencing with high variations in allele frequency.

**Analysis of undesignated possible recombinant samples.** Three of the samples sequenced from the Omicron cohort showed a wide arrange of mutations from both BA.1 and BA.2 variants. These samples appeared in the phylogenetic tree outside of the BA.1 or BA.2 monophyletic clusters (Fig 4). In order to check for recombination, the allelic frequency of the

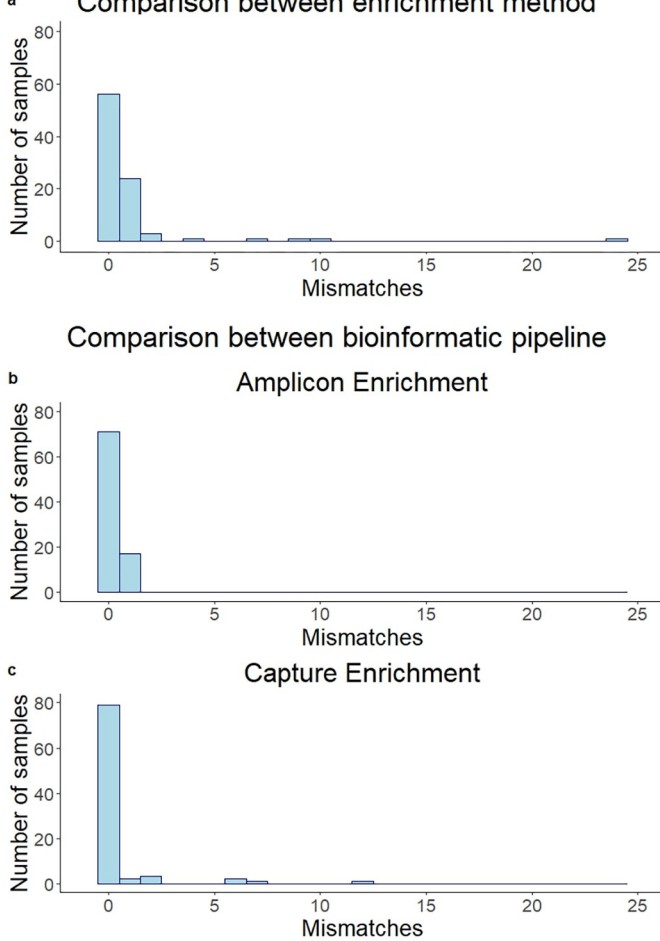

**Fig 5. Histograms of discrepancies in final consensus sequences.** a) Base mismatches in all studied samples using the amplicon-enrichment method or capture-enrichment method. b-c) Discrepancies using two different bioinformatic pipelines, a customised bioinformatic pipeline and Illumina® DRAGEN™ COVID lineage app. b) For ARTIC panel, amplicon-based enrichment. c) For KAPA RNA HyperCap, capture-based enrichment.

defining BA.1 and BA.2 mutations was plotted for these three samples (Fig 6). Allelic frequency showed distinct regional areas of the genome that are highly BA.1 or highly BA.2, suggesting a recombinant sample. Specifically, sample 55 showed two breakpoints: one likely between bases 15240–15714 and other between bases 26060–26530 (Fig 6A and 6B). Samples 57 and 80 shared a single breakpoint likely between bases 26060 and 26530 (Fig 6C–6F).

Sample 55 showed in the CBE a discrepant pattern compared with the amplicon enrichment in the second half of the genome (Fig 6B). This sample showed 9 discrepancies between both methods of enrichment (Fig 5A), and heterogeneous allele frequency. Sample 55 was the only case of Pango designation discrepancy from the analysed samples, with the ABE-generated sequence being declared as B.1.1.529 and the CBE-generated sequence being declared as BA.2 (S2 Table).

## Discussion

We have compared two different methods for viral RNA enrichment in SARS-CoV-2 sequencing. Both methods of enrichment showed differences in base depth, double for the amplicon

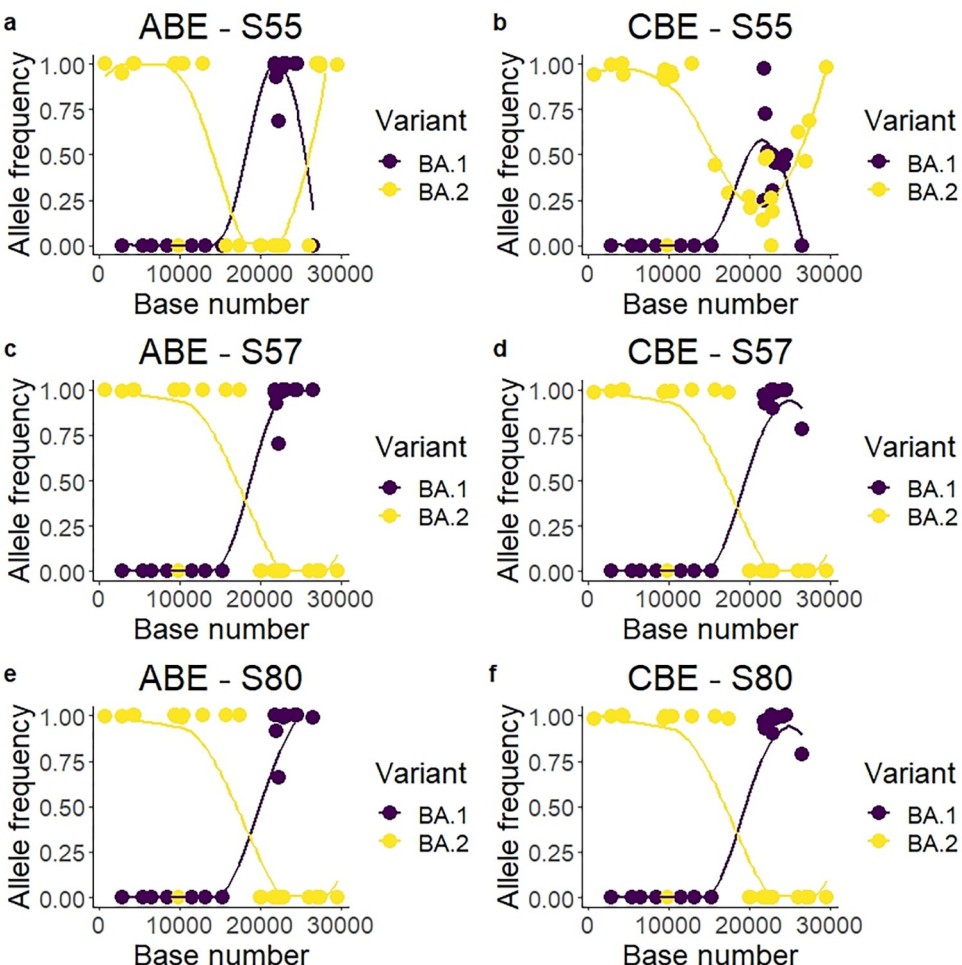

**Fig 6. Variant allele frequency plot for three possible undesignated recombinant samples.** Only BA.1-specific and BA.2-specific mutations were plotted, with the amplicon enrichment (ARTIC panel) on the left (ABE) and the capture enrichment (KAPA RNA HyperCap) on the right (CBE).The high allele frequency, with most SNPs called with over 95% allele frequency, suggesting recombination and not co-infection. Samples 2 and 3 (panels c-f) show likely the same breakpoint, suggesting both coming from the same origin. The lines represent a LOESS local regression for the allele frequency of each variant.

method, although only in the pre-Omicron samples, suggesting that the changes introduced in the amplicon enrichment method in the Omicron samples could have negatively affected the output. Mainly, these changes were the extraction, the automation and the Illumina CovidSeq reagent.

There are currently a few published studies on viral enrichment of SARS-CoV-2 [4, 5, 23–29]. Comparison between studies is complex due to differences in the methods, the analysed variants of SARS-CoV-2, the viral load ranges or the sequencing platforms. Samples per run (expected reads per sample) could be one of the main factors of variability of results across studies. For example, the SNP mismatch frequency detected between enrichment methods in the literature has been reported to be between almost 0 to around 5% [4, 29]. This difference could be likely due to the various factors mentioned above. Notably, as the virus keeps accumulating mutations, genetic diversity increases, and accurate detection of all mismatches becomes more challenging.

## ABE provides higher homogeneity of coverage between samples

The variance across samples was noteworthy, double for the capture method. The amplicon-based method using Illumina reagents relies on normalisation of libraries by tagmentation, assuming equal tagmentation of libraries using the same amount of tagmentation reagent per sample. This proved to be enough to obtain a similar amount of read depth per sample across runs (Fig 1A). In the capture enrichment, libraries are generated first and then non-target DNA is removed, as opposed to amplicon enrichment. This enables the possibility of multiplexing enrichment, increasing the cost-effectiveness of the reagents. In this study we performed a 6-plex library enrichment. This makes individual normalisation of enriched libraries prior to sequencing impossible, causing the observed heterogeneity in read depth (Fig 1A). Singleplex capture is always a possibility and allows for improved normalisation and reads-per-sample homogeneity. However, this also highly increases the cost and hands-on time of the enrichment procedure.

The capture method performed worse in allele frequency. It was found to have more sequences with low allele frequency across the genome. While most samples provided a highly homogeneous allele frequency in SNP detection for both methods of enrichment, a subset of samples processed by the capture method showed a high number of SNPs with low read frequency. The same pattern did not appear in the amplicon method nor in other related samples processed by the capture method, suggesting errors in capture or base calling of these particular samples.

## ABE is sensitive to amplicon loss due to mutations in the primer binding area

In the case of the amplicon enrichment method, we found errors due to the impact of SNPs in primer binding. This enrichment method relies on the proper binding of the primers to specific locations in the genome, so a specific mutation or indel in the primer-binding area can cause a whole amplicon to be missed in the PCR (amplicon dropout) [30, 31]. For this reason, the main disadvantage of the amplicon method in viral sequencing is the need for constant primer update in order to get a panel that is effective for all variants.

Our results showed two cases of primer binding errors causing artefacts in sequencing. In the Delta variant there was a significant drop in reads for amplicon 72 using the ARTIC v3 panel (Fig 2C and 2D). This was not found with our hybrid-capture method, nor for other variants sequenced with the ARTIC panel. Amplicon 72 of the ARTIC panel v3 could be lost in Delta virus sequences due to a Delta-specific deletion 22029–22034 [30, 32]. This deletion overlaps with the primer 72_RIGHT of the ARTIC panel v3 binding area, causing a failure in PCR. This is an example of how an amplicon dropout could cause a loss of NGS reads in specific variants. This phenomenon has already been previously reported by other publications, including the ARTIC consortium. This dropout could also cause the loss in detection of mutation *G21987A* (S:G142D) in Delta, covered by the same amplicon. In this study, 9 samples lacked the mutation, 10 samples left the base as unread (below 10 reads), and only 2 samples included the *G21987A* SNP. Using capture-enrichment, the mutation was detected in all samples, which could be close to the real prevalence [30, 32]. Regarding global data, in March of 2023 (time of writing), 33% of the Delta samples in the GISAID database lacked the *G21987A* mutation [33]. As a consequence, the ARTIC consortium published in June 2021 a v4 primer panel in order to correct for this error. Additionally, we found the mutation *T15521A* in 8 of our 40 Omicron samples, which seems to be another artefact caused by the mispriming of one of the primers included in the ARTIC v4.1 panel. The primer 93_LEFT could hybridise on the amplicon 51 area causing a secondary amplicon. Due to the effect of a mismatch between the

primer and its binding region, mutation *T15521A* seems to have been inserted [34]. This SNP was present in a low allelic frequency for most samples in the amplicon-based enrichment, reinforcing the argument of an artificial insertion (Fig 3A).

The amplicon dropout detected using the ARTIC v3 panel corresponded to the *spike* gene sequence. The spike protein is responsible for the host cell invasion [35], and the target against most designed vaccines. Being the target of vaccines and highly immunogenic, the spike gene is subjected to more selective pressure, as it is driven by evolution to new variants with more immune escape. Therefore, it has a higher mutation rate than the rest of the genome [36]. It is expected that highly mutated areas of the genome such as this one will be more prone to failure in amplification with a primer panel. Given the clinical and epidemiological importance of accurate sequencing of the *spike* gene specifically, frequent update of primer panels is key for an optimal sequencing of new variants. Capture panel probes are unlikely to be affected by SNPs or indels due to most commercial panels consisting of tiled probes of at least 80 bps (120 bps in the case of the KAPA enrichment probes), as it was also found in a recent publication [23]. However, capture probes have been shown to have a reduced yield compared to amplicon enrichment due to the capture of off-target DNA fragments [24].

## Illumina iSeq 100 is sufficient for SARS-CoV-2 sequencing with enrichment

The sequencing system tested in this paper was the Illumina iSeq 100 system, which is the smallest of the Illumina sequencing platforms. While the price per sample of Illumina iSeq is higher compared to the high-capacity sequencing platforms, Illumina iSeq is simpler, easier to install and virtually maintenance-free. In addition, it can provide faster results than other platforms, with a run completed in around 18 hours. We showed that NGS with Illumina iSeq 100 provided over 98% coverage of the SARS-CoV-2 genome in all samples above 50 median reads per base. It is important to note that the amount of reads per base can be optimised by changing the amount of samples per sequencer run. The optimal number of samples per run must therefore be determined by the user depending on the coverage desired and the resources available. A high number of reads per base is ideal as it also allows for detection of small subpopulations within one sample. In the context of the COVID-19 pandemic, this could be useful to detect co-infections with different variants, specially in immunosuppressed patients [37]. Nowadays the number of reads per base could also be relevant in the case of panels of capture enrichment that allow the detection and genotyping of several respiratory viruses in one sample at the same time. The current co-circulation of SARS-CoV-2, influenza and respiratory syncytial virus (RSV) is increasing the demand for detection of co-infection between several respiratory viruses. Hybrid-capture enrichment could be more suited for detecting coinfections between different organisms as it allows for more targets per panel [9].

## Automation of ABE enrichment and library generation provided high-coverage SARS-CoV-2 sequencing

For our amplicon approach, the first cohort was processed manually while for the second one the Hamilton Microlab STAR pipetting platform was used. As the amplicon-based enrichment protocol is generally a simpler protocol than the capture-based one, automation is also easier to program and implement. Manual libraries yielded a higher coverage than automatic ones (Fig 1A, S2 Table), although this could be due to other factors, such as different variants sequenced, different versions of the ARTIC panel used, different NA extraction and amplification reagents and different number of runs per sample. In any case, both systems were successful at genotyping SARS-CoV-2 with a high coverage (Fig 1B). Future users should take into

account that, as automation requires higher reagent volumes, the cost per sample increases using an automatic library preparation pipeline.

## Sample viral load is a determining factor in sequencing success

In order to obtain the most representative data about circulating variants, we usually select for genotyping purposes samples with low-to-mid CT. For this reason, our study focuses on a CT range of 8–26. Samples with low concentration (high CT value) are expected to have a much lower sequencing yield, especially for capture methods [4, 38]. The CT value can be a determining factor for the enrichment method selection, with a previous publication showing amplicon enrichment to be the best-performing enrichment system in low viral load samples [27]. Amplicon enrichment has been previously found to be successful for high genome coverage even at CT values of around 38 [27]. Nevertheless, in our experience, detection of the most challenging mutations with enough quality is increasingly difficult considering the current complexity of the genome of the virus. Every epidemiological service must decide if to opt for a more unbiased approach of sequencing all received samples regardless of CT, despite the risk of lower sequencing quality, or to discard low viral load samples prior to sequencing.

## Different bioinformatic pipelines can cause small differences in the final sequence obtained

This study also compared two different bioinformatic pipelines, an in-house system and Illumina® DRAGEN™ COVID lineage app. Differences in consensus sequence generated by both methods were detected, such as *T15521A* with the ARTIC v4.1 panel, or SNPs with low allele frequency with the capture based enrichment approach. A low allele frequency could cause differences in base calling due to differences in mapping and filtering. A previous publication similarly found differences in base calling when different algorithms were applied [39]. In our cohorts, 80% of the samples processed with the amplicon-based enrichment showed no discrepancies in the sequences generated with different bioinformatic pipelines, and only one discrepancy was found for the other samples. For the capture-based enrichment, 90% of the sequences were identical. Both systems proved to be sufficient for analysis and consensus sequence determination, and Illumina DRAGEN is an user-friendly system that could be implemented in any sequencing facility.

## Limitations and potential improvements

There were limitations to our research. As we focused on previously-confirmed positive samples with low-to-mid CT value, samples were only sequenced once per method with no replicates, and no negative or positive controls were added in the runs. We encourage epidemiological vigilance services to add a negative and a positive control in diagnostic routine in order to discard false positives due to contamination as well as to discard potential sequencing errors.

Despite this, the level of agreement across samples was high using different enrichment methods and bioinformatic analysis. Both methods were also able to detect likely recombinant samples, although it must be noted that the ECDC recommends 1000 bps sequencing read length in order to detect recombinant samples [19], more than the 150 bps allowed by Illumina iSeq 100.

A future, more detailed analysis of SARS-CoV-2 sequencing could include new circulating variants, coinfections, other input samples such as saliva, different sample transport mediums, or different RNA extraction procedures, library preparation kits and sequencers. Nonetheless, our study provides insight on the quality of the Illumina iSeq 100 as a sequencing instrument

for SARS-CoV-2, as well as the pros and cons of the two main mechanisms for viral RNA enrichment, using two of the most common market-available library platforms. Illumina iSeq is economical and virtually maintenance-free, and therefore its implementation should be easier than other possible options.

## Final remarks

In summary, both enrichment methods showed a high sequencing quality in the samples studied. Nevertheless, capture enrichment with the KAPA RNA Hypercap reagent increased the difficulty of an optimal library normalisation and therefore provided an uneven number of reads when multiplexing samples in the same sequencing run. On the other hand, the ARTIC amplicon-based enrichment was sensitive to SNPs as well as to deletions. These SNPs could cause a decrease in primer binding efficiency and a sequencing bias in which samples from certain variants had a deeper sequencing than others. Constant updates of the primer panels are therefore required to avoid amplicon dropouts.

In this study we showed the behaviour of two SARS-CoV-2 genotyping methods with a wide variety of variants considered as VOC throughout the pandemic, including the Omicron variant. In this way, we can say that concordant results were obtained for variants Alpha, Beta, Gamma, Delta, Zeta, Iota, Mu, Omicron and even possible recombinant genomes. Another important point is that we showed the performance of the automatization of the library preparation and an app for bioinformatic analysis of NGS data that could simplify the overall genotyping process.

## Supporting information

**S1 File. Custom protocol for RT-PCR amplification of SARS-CoV-2 RNA.**
(DOCX)

**S1 Table. ARTIC primers (v3 & v4.1) used for amplicon-based enrichment.**
(XLSX)

**S2 Table. Coverage and depth per sample using all enrichment methods and bioinformatic analysis.**
(XLSX)

**S1 Dataset.**
(DOCX)

## Acknowledgments

We would like to acknowledge the staff from the Microbiology service of the Complexo Hospitalario Universitario de Vigo (CHUVI), for their contribution to epidemiological surveillance, their work and dedication.

## Author Contributions

**Conceptualization:** Benito Regueiro-García.

**Formal analysis:** Sonia Pérez.

**Funding acquisition:** Benito Regueiro-García.

**Investigation:** Carlos Daviña-Núñez, Anniris Rincón-Quintero, Ana Treinta-Álvarez, Montse Godoy-Diz.

**Methodology:** Carlos Daviña-Núñez, Anniris Rincón-Quintero, Ana Treinta-Álvarez, Montse Godoy-Diz.

**Resources:** Benito Regueiro-García.

**Software:** Sonia Pérez.

**Supervision:** Sonia Pérez, Silvia Suárez-Luque.

**Validation:** Jorge Julio Cabrera-Alvargonzález, Silvia Suárez-Luque.

**Visualization:** Carlos Daviña-Núñez, Jorge Julio Cabrera-Alvargonzález.

**Writing – original draft:** Carlos Daviña-Núñez.

**Writing – review & editing:** Sonia Pérez, Jorge Julio Cabrera-Alvargonzález, Anniris Rincón-Quintero, Benito Regueiro-García.

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
