## [Decision Letter · Decision Letter 0]

22 Jan 2024

PONE-D-23-21634Performance of amplicon and capture based next-generation sequencing approaches for the epidemiological surveillance of Omicron SARS-CoV-2 and other variants of concern.PLOS ONE

Dear Dr. Perez,

Thank you for submitting your manuscript to PLOS ONE. After careful consideration, we feel that it has merit but does not fully meet PLOS ONE’s publication criteria as it currently stands. Therefore, we invite you to submit a revised version of the manuscript that addresses the points raised during the review process.

We look forward to receiving your revised manuscript.

Kind regards,

Hin Fung Tsang

Academic Editor

PLOS ONE

Journal Requirements:

Reviewers' comments:

Reviewer's Responses to Questions

**Comments to the Author**

1. Is the manuscript technically sound, and do the data support the conclusions?

Reviewer #1: Yes

Reviewer #2: Yes

Reviewer #3: Yes

2. Has the statistical analysis been performed appropriately and rigorously? 

Reviewer #1: Yes

Reviewer #2: Yes

Reviewer #3: Yes

3. Have the authors made all data underlying the findings in their manuscript fully available?

Reviewer #1: Yes

Reviewer #2: No

Reviewer #3: Yes

4. Is the manuscript presented in an intelligible fashion and written in standard English?

Reviewer #1: Yes

Reviewer #2: Yes

Reviewer #3: Yes

5. Review Comments to the Author

Reviewer #1: 1. Experimental Setup:

Two bioinformatic pipelines were used to examine the samples in the study, and Illumina iSeq 100 was used for sequencing. The results are more reliable when several methods and pipelines are used.

2. Challenges with Amplicon Method:

The paper points out drawbacks of the amplicon enrichment technique, such as amplicon dropout and mispriming, particularly in the context of particular mutations like G21987A and T15521A. This emphasizes the necessity of ongoing primer modifications for efficient sequencing.

Reviewer #2: I would recommend the following suggestions

1) Kindly arrange the manuscript sections as per journal format. Read the journal guidelines for manuscript.

2) I would suggest to include thermal profile for RT-PCR in tabular form.

3) The primer sequences should be made available as supplementary data.

4) What you meant by "selective pressure", line 340.

5) I would like the authors to highlight why they have not included positive or negative controls in their investigation.

6) I would suggest to add a separate heading for challenges at each step, faced to sequence SARS-CoV-2: that they mentions throughout this manuscript.

Reviewer #3: This scientific article is suitable for publication because it contains an ethical license to conduct research, and it also clarifies and details precise details, while proving them with diagrams and curves.

6. PLOS authors have the option to publish the peer review history of their article (what does this mean?). If published, this will include your full peer review and any attached files.

Reviewer #1: **Yes: **Elhadi Abdalla Ahmed

Reviewer #2: No

Reviewer #3: **Yes: **Murtadha Abbas

---

## [Author Response · Author response to Decision Letter 0]

11 Mar 2024

We want to thank all reviewers for their time, for their comments and for their reviews. We appreciate the feedback. Please find our response to your comments below.

Reviewer 1

1. Experimental Setup:

Two bioinformatic pipelines were used to examine the samples in the study, and Illumina iSeq 100 was used for sequencing. The results are more reliable when several methods and pipelines are used.

Our aim for this paper was to show the feasibility of the sequencing of SARS-CoV-2. By showing the effectivity of a user-friendly pipeline (Illumina DRAGEN) and an economic sequencing platform (Illumina iSeq 100), we try to encourage researchers to join the global epidemiological vigilance of SARS-CoV-2 and other respiratory viruses, even when resources or bioinformatic capacity is limited in the facility.

We agree that several pipelines and sequencing platforms can ensure solid results. In order to have a more robust analysis, we compared two enrichment methods and bioinformatic pipelines.

2. Challenges with Amplicon Method:

The paper points out drawbacks of the amplicon enrichment technique, such as amplicon dropout and mispriming, particularly in the context of particular mutations like G21987A and T15521A. This emphasizes the necessity of ongoing primer modifications for efficient sequencing.

We agree that amplicon panels need constant updates, especially in the case of RNA viruses with high mutation rate, as mentioned in the discussion section “ABE is sensitive to amplicon loss due to mutations in the primer binding area” (lines 324-371) . Thank you for your comment.

Reviewer 2

Reviewer #2: I would recommend the following suggestions

1) Kindly arrange the manuscript sections as per journal format. Read the journal guidelines for manuscript.

Thank you for the comment. We tried to arrange the sections according to journal format (3 subsections in fonts 18, 16 and 14 respectively). Main headings were for the main sections (e.g. Introduction, Methods, Results and Discussion). Subsections were used across the text, and another sublevel was used only in the Methods section, subsection “Sequencing approaches” (lines 89, 101 and 109).

In order to make the manuscript more easy to follow, and according to your suggestion 6), we have also added subsections in the discussion section of the manuscript. Additionally, one paragraph has been moved above (lines 292-301) to improve the order of the arguments exposed.

If there is something else we have missed, please let us know and we will correct it accordingly.

2) I would suggest to include thermal profile for RT-PCR in tabular form.

Thank you for the suggestion. Upon your comment, we considered that the RT-PCR protocol was perhaps scarce in detail so we have added a supplementary file (S1 file) with the full protocol for the manual amplification. In the automatic amplification, as it was performed with the Illumina CovidSeq protocol without changes, we have added the reference of the Illumina Reference Guide that was followed (line 105), as we believe it provides enough detail for another user to complete the protocol as intended.

3) The primer sequences should be made available as supplementary data.

Thank you for pointing this out. We have added another supplementary table (S1 Table) that includes all primers from both versions of ARTIC used.

4) What you meant by "selective pressure", line 340.

We mean selective pressure as the cause for a genotype to change in order to adapt. The spike protein is highly immunogenic and the target of vaccines. For this reason, evolution forces new variants of this gene in the population, in order to keep escaping the host’s immune system. This has been shown in publications such as this one: 

Amicone, M., Borges, V., Alves, M.J., Isidro, J., Zé-Zé, L., Duarte, S., Vieira, L., Guiomar, R., Gomes, J.P. and Gordo, I., 2022. Mutation rate of SARS-CoV-2 and emergence of mutators during experimental evolution. Evolution, medicine, and public health, 10(1), pp.142-155.

In the context of our publication, additional mutations mean more likelihood of a mutation appearing in the primer-binding area, affecting amplicon homogeneity by drop in primer binding.

We have added an additional explanation in the manuscript (lines 359-364). The publication above has also been added to the bibliography as citation number 36.

5) I would like the authors to highlight why they have not included positive or negative controls in their investigation.

Our study focuses on comparing two kits commonly used in routine diagnostics in our facility, and we believed that an efficient comparison in methodology in performance could be achieved by just focusing on the analysis of clinical samples. This also allowed us to focus on as many SARS-CoV-2 variants as possible in order to show potential defects in the enrichment across all the variability circulating. More controls are definitely an improvement in the methodology and we encourage their use in common diagnostic practice. In order to clarify this for the scientific community, we have added a paragraph detailing this (lines 450-455).

6) I would suggest to add a separate heading for challenges at each step, faced to sequence SARS-CoV-2: that they mentions throughout this manuscript.

The discussion section has been structured according to subsections in order to discuss each challenge from the different approaches (viral load, enrichment method, sequencing platform or bioinformatic analysis amongst others). Thank you for the recommendation.

Reviewer 3

Reviewer #3: This scientific article is suitable for publication because it contains an ethical license to conduct research, and it also clarifies and details precise details, while proving them with diagrams and curves.

Thank you very much for your time, for your comments and for your review. We appreciate the feedback and your kind words.

---

## [Editor Report · Decision Letter 1]

15 Mar 2024

Performance of amplicon and capture based next-generation sequencing approaches for the epidemiological surveillance of Omicron SARS-CoV-2 and other variants of concern.

PONE-D-23-21634R1

Dear Dr. Sonia Perez,

We’re pleased to inform you that your manuscript has been judged scientifically suitable for publication and will be formally accepted for publication once it meets all outstanding technical requirements.

Kind regards,

Hin Fung Tsang

Academic Editor

PLOS ONE